# Learn from A Rationalist: Distilling Intermediate Interpretable Rationales

**Jiayi Dai** [1 2]    **Randy Goebel** [1 2]

## Abstract

Because of the pervasive use of deep neural networks (DNNs), especially in high-stakes domains, the interpretability of DNNs has received increased attention. The general idea of rationale extraction (RE) is to provide an interpretable-by-design framework for DNNs via a select-predict architecture where two neural networks learn jointly to perform feature selection and prediction, respectively. Given only the remote supervision from the final task prediction, the process of learning to select subsets of features (or *rationales*) requires searching in the space of all possible feature combinations, which is computationally challenging and even harder when the base neural networks are not sufficiently capable. To improve the predictive performance of RE models that are based on less capable or smaller neural networks (i.e., the students), we propose **REKD** (**R**ationale **E**xtraction with **K**nowledge **D**istillation) where a student RE model learns from the rationales and predictions of a teacher (i.e., a *rationalist*) in addition to the student's own RE optimization. This structural adjustment to RE aligns well with how humans could learn effectively from interpretable and verifiable knowledge. Because of the neural-model agnostic nature of the method, any black-box neural network could be integrated as a backbone model. To demonstrate the viability of REKD, we conduct experiments with multiple variants of BERT and vision transformer (ViT) models. Our experiments across language and vision classification datasets (i.e., IMDB movie reviews, CIFAR 10 and CIFAR 100) show that REKD significantly improves the predictive performance of the student RE models. The code is publicly available: https://github.com/JiayiDai/REKD.

## 1. Introduction

*"If I have seen further, it is by standing on the shoulders of giants."*
                                                                    — Isaac Newton

Large deep neural networks (DNNs), including ResNet (He et al., 2015), BERT (Devlin et al., 2019), GPTs (Brown et al., 2020; OpenAI, 2024) and vision transformer (ViT) (Dosovitskiy et al., 2021), have achieved superior predictive performance across language and vision tasks. However, their knowledge is represented in an opaque way (i.e., black-box), which raises concerns about their applications in high-stakes domains, such as healthcare (Dai et al., 2025; Fahad et al., 2025) and finance (Cao, 2022; Mienye et al., 2024). To improve the trustworthiness of black-box models, a large volume of research on explainable artificial intelligence (XAI) has emerged (Ali et al., 2023; Longo et al., 2024). The most popular XAI tools to date have been post-hoc or plug-and-play methods, such as LIME (Ribeiro et al., 2016), SHAP (Lundberg & Lee, 2017), Integrated Gradients (Sundararajan et al., 2017) and Grad-CAM (Selvaraju et al., 2017) due to their convenience, but they do not ensure *faithful* explanations, i.e., the features identified as important may not truly be important for a model's prediction, which could be misleading and potentially harmful.

Rationale extraction (RE), proposed by Lei et al. (2016), offers an interpretable-by-design alternative through a select-predict architecture where two neural networks, i.e., a generator and a predictor, learn jointly to generate a rationale and make a prediction based on the rationale. The design guarantees that the selected features in the rationale are actually the features used for the prediction (i.e., faithful). However, with only the remote supervision signal from the final prediction, the training process is difficult: the generator relies on the guidance of the predictor to select important features while the predictor relies on the output of the generator to learn task prediction. This "chicken and egg" problem is significantly exacerbated when the base neural networks are not sufficiently capable, which leads to the main research question of this paper: *how to improve the RE models that are based on less capable or smaller neural networks?*

We approach the question by first recalling that this "chicken and egg" phenomenon is very common in human scientific activities. Consider the challenge of physics law discovery, such as deriving Newton's law of universal gravitation from

[1]Department of Computing Science, University of Alberta, Edmonton, Canada [2]Alberta Machine Intelligence Institute, Edmonton, Canada. Correspondence to: Jiayi Dai <dai1@ualberta.ca>.

*Proceedings of the 43rd International Conference on Machine Learning*, Seoul, South Korea. PMLR 306, 2026. Copyright 2026 by the author(s).

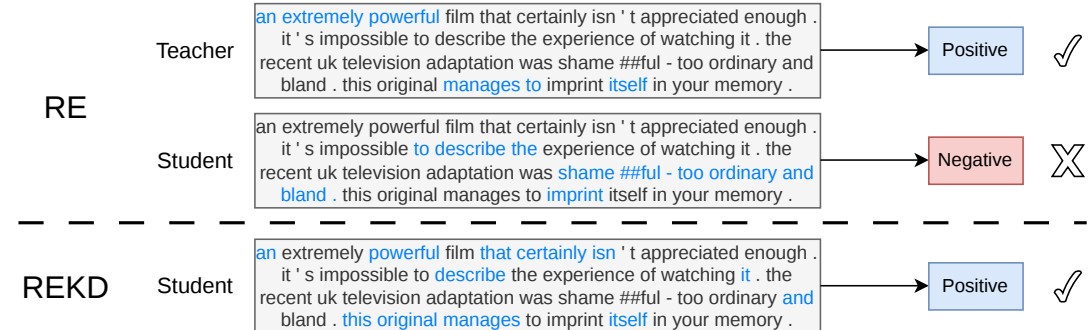

*Figure 1.* An actual example of REKD from IMDB movie reviews. Rationales are highlighted in blue. The student RE model (BERT Mini@RE) originally gets confused by the negative description of a TV adaption of the movie and makes a wrong prediction, while its REKD version learns from the rationale of the teacher (BERT Base@RE) to predict accurately.

scratch. One cannot derive the precise mathematical formula (the predictor) without first identifying that mass and distance are the critical governing variables (the rationale). Conversely, it is difficult to isolate these specific variables from the myriad of irrelevant factors (e.g., color, temperature, one's religious beliefs) without a working formula to verify their predictive power. However, once Isaac Newton published the gravity theory in 1687 (Newton, 1999), which was written in an interpretable and verifiable way, even an average human could utilize the knowledge to make predictions on gravity force almost as accurately as Newton himself. In RE, if the teacher's rationales have been verified to be more powerful in prediction than the students', we should expect that the students could be improved by following the teacher. Also, because the feature selection layer serves as a neural-model agnostic interface, the knowledge on what features are important can be distilled from a strong teacher (e.g., Isaac Newton) to students (e.g., average humans) regardless of their underlying neural architectures.

Motivated by this intuition, we propose **REKD** (**R**ationale **E**xtraction with **K**nowledge **D**istillation), a framework that enables RE students to learn robust feature selections and predictions from a teacher's supervision in addition to the students' RE exploration. We employ the Straight-Through Gumbel-Softmax estimator (Jang et al., 2017) to make the feature selection process of RE differentiable, which requires a temperature annealing process for gradient stability and sampling accuracy. By synchronizing the knowledge distillation temperature with the annealing scheduler, we impose a progressive complexity curriculum: the student initially absorbs broad knowledge from the teacher's softened Gumbel-Softmax distributions to guide exploration, and gradually transitions to mimicking sharp, high-confidence feature selections as the temperature anneals, enforcing precision during the discretization phase.

We summarize the three key contributions of the paper:

1. We identify the "chicken and egg" dilemma in training the select-predict architecture of RE, illustrating why lightweight students could struggle to learn effective rationales and achieve good predictive performance without external guidance.

2. We propose REKD, a neural-model agnostic distillation framework for rationale extraction that leverages the intrinsic curriculum of the Gumbel-Softmax annealing.

3. We validate our approach on both language (IMDB movie reviews) and vision (CIFAR 10/100) classification tasks using multiple variants of BERT and ViT as RE backbones. Experiments demonstrate that REKD significantly improves the predictive performance of the student RE models.

## 2. Methods

In this section, we first describe the two key components of REKD, i.e., rationale extraction (RE) with Straight-Through Gumbel-Softmax and knowledge distillation (KD), and then we introduce REKD by unifying RE and KD via a shared temperature annealing scheduler.

### 2.1. Rationale Extraction via Straight-Through Gumbel-Softmax

Following the original design by Lei et al. (2016), the rationale extraction framework has two neural components, a generator and a predictor, to separately perform rationale generation and classification tasks. In this section, we formalize the framework and also incorporate Straight-Through Gumbel-Softmax (Jang et al., 2017) for differentiable rationale sampling. We use a one-sample input (instead of a batch of samples) and a bold font for vectors in the writing for clarity.

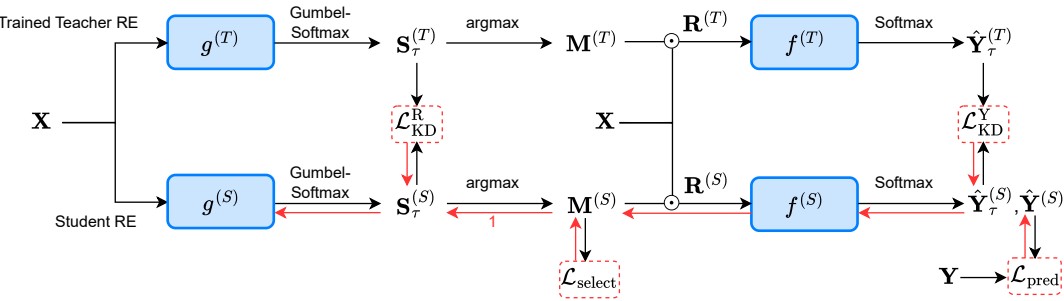

*Figure 2.* The schematic of REKD. The student RE model learns from the teacher's supervision on feature selections and predictions in addition to its own RE exploration via Straight-Through Gumbel-Softmax. The red arrows indicate the gradient flows and the red 1 represents the Straight-Through estimation, i.e., $\frac{\partial \mathbf{M}}{\partial \mathbf{S}} \approx 1$. The definitions for the computation and the variables are in Section 2.

**Generator**  Let the input vector be denoted by $\mathbf{X} \in \mathbb{R}^{L \times D}$, where $L$ is the sequence length or the number of features and $D$ is the embedding dimension. We define the generator neural network as $g(\cdot)$, parameterized by $\theta_g$, which maps the input to unnormalized logits $\mathbf{Z} \in \mathbb{R}^{L \times 2}$ corresponding to Bernoulli distributions representing feature selections.

Given the logits $\mathbf{Z}$ from the generator, the standard continuous Gumbel-Softmax samples $\mathbf{S} \in \Delta_1^L$ are computed as

$$S_{l,i} = \frac{\exp((Z_{l,i} + G_{l,i})/\tau)}{\sum_{j=0}^{1} \exp((Z_{l,j} + G_{l,j})/\tau)} \qquad (1)$$

where $l$ denotes the feature index, $G_{l,i}$ are i.i.d samples from Gumbel$(0, 1)$[1] and $\tau$ is the temperature that controls the "sharpness" of the distribution. By discretizing the soft samples $\mathbf{S}$, we obtain a binary mask $\mathbf{M} \in \{0, 1\}^L$ over all features

$$M_l = \underset{i \in \{0,1\}}{\arg\max}(S_{l,i}) \qquad (2)$$

where index 1 corresponds to the "selected" class. Thus, $M_l = 1$ indicates the $l^{\text{th}}$ feature is selected; otherwise, it is not selected. During the backward pass, the argmax is approximated as an identity function and we estimate the gradient with $\nabla_{\theta_g}\mathbf{M} \approx \nabla_{\theta_g}\mathbf{S}$, i.e., Straight-Through (Bengio et al., 2013).

Given the mask $\mathbf{M}$ and the original input $\mathbf{X}$, a rationale $\mathbf{R}$ is then defined as $(\mathbf{M} \odot \mathbf{X}) \in \mathbb{R}^{L \times D}$, i.e., a subset of features obtained by applying the mask over the input. The unselected features are replaced by zero vectors, which guarantees that only the selected features are utilized for the final prediction.

Then, we define a selection loss term[2] as the (mean) squared

loss between the generated rationale length and a target rationale length

$$\mathcal{L}_{\text{select}} = \left(\sum_{l=0}^{L-1} M_l - L \cdot p_{\text{target}}\right)^2 \qquad (3)$$

where $p_{\text{target}} \in [0, 1]$ is hyperparameter representing a desired rationale ratio over the full input length $L$. It effectively encourages the rationale length to be close to the target.

**Predictor**  The predictor network, denoted as $f(\cdot)$, receives the rationale $\mathbf{R} = \mathbf{M} \odot \mathbf{X}$ and outputs unnormalized logits $\mathbf{Q} = f(\mathbf{M} \odot \mathbf{X})$. Then, the final prediction probability distribution $\hat{\mathbf{Y}}$ for a C-way classification are obtained via a softmax function, $\hat{\mathbf{Y}} = \text{softmax}(\mathbf{Q})$. The network is trained to minimize the cross-entropy loss between $\hat{\mathbf{Y}}$ and the ground truth label $\mathbf{Y}$:

$$\mathcal{L}_{\text{task}} = \text{CE}(\hat{\mathbf{Y}}, \mathbf{Y}) = -\sum_{c=0}^{C-1} Y_c \log(\hat{Y}_c) \qquad (4)$$

**Joint learning objective**  The overall objective function for rationale extraction is defined by combining the selection loss in Equation (3) and the prediction loss in Equation (4)

$$\begin{aligned} \mathcal{L}_{\text{RE}} &= \mathcal{L}_{\text{pred}} + \lambda_{\text{select}}\mathcal{L}_{\text{select}} \\ &= \text{CE}(\hat{\mathbf{Y}}, \mathbf{Y}) + \lambda_{\text{select}}\left(\sum_{l=0}^{L-1} M_l - L \cdot p_{\text{target}}\right)^2 \end{aligned} \qquad (5)$$

where $\lambda_{\text{select}}$ controls the strength of the selection regularization. The prediction loss guides the predictor and also encourages the generator to select the features that are important for making accurate predictions (i.e., sufficiency); the selection loss (usually with a small target ratio $p_{\text{target}}$) helps to avoid selecting not or less important features (i.e.,

---

[1]In implementation, as suggested by Jang et al. (2017), we obtain $G_{l,i} = -\ln(-\ln(U_{l,i}))$ with $U_{l,i} \sim \text{Uniform}(0, 1)$.

[2]Some earlier work (Lei et al., 2016; Bastings et al., 2019) used a $L_0$ loss that complicates rationale length control; later Jain et al. (2020) & Paranjape et al. (2020) modified it to a rectifier loss that

induces a penalty only when a target length is passed.

necessity). Overall, the joint objective encourages the sufficiency and necessity of the selected features as the premises for accurate task predictions.

## 2.2. Distilling Rationale Extraction

Knowledge distillation (KD) (Hinton et al., 2015) was proposed to improve the predictive performance of lightweight neural models by transferring knowledge from a teacher model. In REKD, the goal of KD is to let a teacher RE model deliver this piece of knowledge to a student: "Given input $\mathbf{X}$, these features $\mathbf{R}^{(T)}$ are important and, because of these features, the final prediction is $\hat{\mathbf{Y}}^{(T)}$". It means the KD loss function should encourage the student generator and predictor to learn from the rationales and task predictions of the teacher. We use superscripts $(T)$ for Teacher and $(S)$ for Student. Subscript $\tau$ is added to indicate that a probability distribution is temperature-scaled by $\tau$ for better clarity in KD-related writing. We then define the two loss terms for distilling rationales and task predictions, respectively.

**Generator distillation** Rationale distillation is performed by matching the Gumbel-Softmax distributions defined by Equation (1) from the student and the teacher generators across all $L$ features. To be specific, for each feature $l$, we compute the Kullback–Leibler divergence ($D_{\text{KL}}$) between $\mathbf{S}_{\tau,l}^{(S)}$ and $\mathbf{S}_{\tau,l}^{(T)}$. Formally, the rationale distillation loss is defined as

$$
\begin{aligned}
\mathcal{L}_{\text{KD}}^{\text{R}} &= \sum_{l=0}^{L-1} D_{\text{KL}}(\mathbf{S}_{\tau,l}^{(T)} \parallel \mathbf{S}_{\tau,l}^{(S)}) \\
&= \sum_{l=0}^{L-1} \sum_{i=0}^{1} S_{\tau,l,i}^{(T)} \log \left( \frac{S_{\tau,l,i}^{(T)}}{S_{\tau,l,i}^{(S)}} \right)
\end{aligned}
\tag{6}
$$

**Predictor distillation** Distilling a predictor is to match the student's and the teacher's predicted class probability distributions that are scaled by $\tau$, i.e., $\hat{\mathbf{Y}}_{\tau}^{(S)}$ and $\hat{\mathbf{Y}}_{\tau}^{(T)}$, which is the same as in standard knowledge distillation (Hinton et al., 2015). Given the predictor logits $\mathbf{Q}^{(T)}$ and $\mathbf{Q}^{(S)}$, we have the $D_{\text{KL}}$ loss defined as

$$
\begin{aligned}
\mathcal{L}_{\text{KD}}^{\text{Y}} &= D_{\text{KL}}(\hat{\mathbf{Y}}_{\tau}^{(T)} \parallel \hat{\mathbf{Y}}_{\tau}^{(S)}) \\
&= D_{\text{KL}} \left( \text{softmax} \left( \frac{\mathbf{Q}^{(T)}}{\tau} \right) \parallel \text{softmax} \left( \frac{\mathbf{Q}^{(S)}}{\tau} \right) \right)
\end{aligned}
\tag{7}
$$

**Knowledge distillation loss** The overall knowledge distillation loss is defined below by combining the generator and the predictor distillation loss terms of Equation (6) and (7)

$$
\begin{aligned}
\mathcal{L}_{\text{KD}} &= \lambda_R \mathcal{L}_{\text{KD}}^{\text{R}} + \tau^2 \mathcal{L}_{\text{KD}}^{\text{Y}} \\
&= \lambda_R \sum_{l=0}^{L-1} D_{\text{KL}}(\mathbf{S}_{\tau,l}^{(T)} \parallel \mathbf{S}_{\tau,l}^{(S)}) + \tau^2 D_{\text{KL}} \left( \mathbf{Y}_{\tau}^{(T)} \parallel \mathbf{Y}_{\tau}^{(S)} \right)
\end{aligned}
\tag{8}
$$

where the constant $\lambda_R$ is a hyperparameter controlling the weight of generator distillation. In standard KD with a high temperature $\tau$, the KD loss term should be scaled by $\tau^2$ to balance the scales of the KD and the original task loss terms because the gradient magnitude is scaled by $1/\tau^2$ when scaling the logits by $1/\tau$ (Hinton et al., 2015). Hence, we scale the $\mathcal{L}_{\text{KD}}^{\text{Y}}$ term by $\tau^2$. However, it is not required when aligning the Gumbel-Softmax distributions in $\mathcal{L}_{\text{KD}}^{\text{R}}$ as the Gumbel-Softmax estimator handles the gradient scaling internally when approximating the discrete feature selections with $\tau \to 0$ (See more discussion in Appendix B).

## 2.3. Rationale Extraction with Knowledge Distillation

Finally, we introduce REKD where a student RE model learns by distilling a teacher's rationales and predictions (Section 2.2) in addition to the autonomous exploration via the select-predict architecture of RE (Section 2.1).

**Temperature scheduler** We define an exponential annealing scheduler for temperature $\tau_k$ (denoting $\tau$ at step $k$) that decreases with a decay rate $\gamma$ over $K$ optimization steps from an initial high temperature $\tau_0$ to a final low temperature $\tau_K$

$$
\tau_k = \tau_0 e^{-\gamma k} \quad \text{where} \quad \gamma = \frac{1}{K} \log \left( \frac{\tau_0}{\tau_K} \right)
\tag{9}
$$

Such a temperature annealing scheduler is a common practice with the Gumbel-Softmax estimator. It helps to ease the trade-off between maintaining low-variance gradients (at large $\tau$) and achieving accurate discrete approximations (at small $\tau$) (Jang et al., 2017).

**REKD objective** The REKD objective is to provide the student with the teacher's robust feature selection and prediction knowledge while still allowing the student to learn from its own experience of rationale extraction exploration. Then, the REKD training objective is defined below as a weighted combination of $\mathcal{L}_{\text{RE}}$ in Equation (5) and $\mathcal{L}_{\text{KD}}$ in Equation (8)

$$
\begin{aligned}
\mathcal{L}_{\text{REKD}} &= \alpha \mathcal{L}_{\text{RE}} + (1-\alpha) \mathcal{L}_{\text{KD}} \\
&= \alpha (\mathcal{L}_{\text{pred}} + \lambda_{\text{select}} \mathcal{L}_{\text{select}}) \\
&\quad + (1-\alpha)(\lambda_R \mathcal{L}_{\text{KD}}^{\text{R}} + \tau^2 \mathcal{L}_{\text{KD}}^{\text{Y}})
\end{aligned}
\tag{10}
$$

where $\alpha \in [0, 1]$ is a hyperparameter that weighs RE and KD loss terms.

The design of the REKD objective function provides the student model with teacher guidance along with its own RE exploration. By synchronizing the KD temperature with the annealing scheduler of $\tau$, the loss also implicitly creates a *curriculum learning* strategy for KD where the student gradually transitions from initially learning the soft knowledge of the teacher (i.e., less sharp Gumbel-Softmax distributions at high $\tau$) to later learning the hard feature selections as $\tau$ decays.

## 3. Experimental Setup

In this section, we describe the details on the used base neural networks, the datasets, the training details and the evaluation metrics.

### 3.1. Base Neural Networks

Because of the neural-model agnostic nature of both RE and the distillation process, any black-box neural network could be implemented as the base generator or predictor model for the REKD experiments. In our implementation, the BERT variants, *bert-base-uncased* (Devlin et al., 2019) as a teacher and *bert-small* and *bert-mini* (Turc et al., 2019) as students, have been used for the language task, IMDB movie reviews. The ViT variants, *vit-base-patch16-224* (Dosovitskiy et al., 2021) as a teacher and *vit-small-patch16-224* and *vit-tiny-patch16-224* (Touvron et al., 2021) as students, have been used for the vision tasks, CIFAR 10 and CIFAR 100. Each type of base neural model is used for both the generator and the predictor to form one RE model. All ViTs and BERTs are used starting with their pretrained weights.

**Generator** When a ViT or BERT is used as the generator network, after its forward pass, we obtain the contextual-aware representation vector for each patch/token feature and then we map each feature representation vector to a Gumbel-Softmax distribution for feature selection. The dimension of the feature representation varies depending on the hidden dimension of the neural architecture, but the feature selection distribution has a fixed dimension of 2 (i.e., probabilities for not selected vs. selected).

**Predictor** The predictor performs as a usual classifier. However, in implementation for both ViTs and BERTs, to zero out the unselected features, we explicitly obtain the patch/token embeddings of the inputs and apply the masks over the embedding vectors. Then, the masked embeddings are given to the rest of the networks, such as adding the CLS tokens and positional embeddings.

### 3.2. Datasets

**CIFAR 10/100** The vision part has two datasets: CIFAR 10 and CIFAR 100 (Krizhevsky, 2009). CIFAR 10 originally

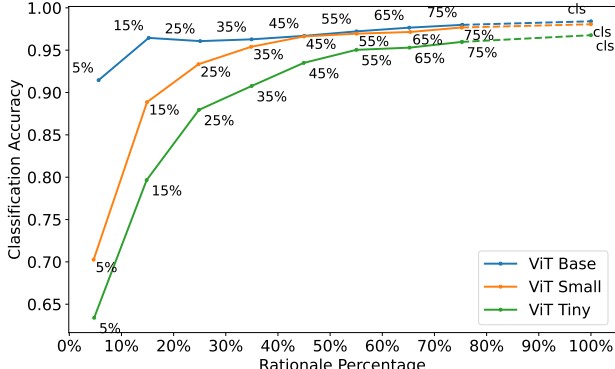

*Figure 3.* Rationale ratio vs. accuracy for ViT Base, Small and Tiny on CIFAR 10 by varying $p_{\text{target}}$ from 5% to 75%. The x-axis is the actual rationale ratio and the y-axis is the accuracy. The target rationale ratio $p_{\text{target}}$ is used as the label for each data point. The label "cls" represents the classification result. Each data point is an average of 10 runs. See Table 11 for the detailed numbers.

consists of 60,000 images (50,000 for training and 10,000 for testing) with totally 10 balanced classes. CIFAR 100 also has 60,000 images with the same split portions (50,000 training & 10,000 testing) but has 100 balanced classes. We randomly split the original training portion of CIFAR 10/100 to 40,000 (80%) for actual training and 10,000 (20%) for developing. Every split has perfectly balanced classes. For CIFAR 100, we use the 20 "coarse" classes instead of 100 classes. All images are upscaled from a resolution of 32×32 to 224×224 to fit the ViT models. Because each patch of the ViTs is of size 16×16 (Dosovitskiy et al., 2021). We have a total of 196 (=224×224/(16×16)) features for each sample.

**IMDB** The IMDB movie review dataset (Maas et al., 2011) is a binary classification dataset for sentiment analysis. It originally has 25,000 training and 25,000 testing samples. We randomly divide the original training split into 20,000 (80%) for training and 5,000 (20%) for developing. Each split has perfectly balanced classes. Each sample is truncated or padded to 256 tokens/features after tokenization.

### 3.3. Training Setup

We used 35 epochs for RE and REKD and 20 epochs for classification, which were sufficient for convergence. During the total training epochs, the version that achieved the lowest developing loss ($\mathcal{L}_{\text{RE}}$ for RE and REKD[3]; $\mathcal{L}_{\text{pred}}$ for classification) was saved as the final model. For RE and REKD, an initial temperature of $\tau_0 = 5$ and a final temperature of $\tau_K = 0.1$ were used for the annealing scheduler, i.e.,

---

[3]Note that it is important to use $\mathcal{L}_{\text{RE}}$, not the whole loss, as the REKD developing loss metric for determining the best version.

Equation (9), which adjusted $\tau$ every 100 training steps. For all neural models, the learning rate was 1e-5 and the batch size was 32. For RE and REKD models, the target selection sparsity $p_{\text{target}}$ was 15% for ViTs on CIFAR 10/100 and 10% for BERTs on IMDB (except for BERT Small@RE which used $p_{\text{target}} = 9.5\%$) and the rationale distillation weight $\lambda_R$ was 0.5. All experiments were repeated for 10 runs over 10 distinct random seeds = [2026, 2027, ..., 2035]. In REKD, for each task, the teacher model of one particular seed was used for all its student models' repeated runs.

The loss function described in Equation (10) introduces two important hyperparameters $\lambda_{\text{select}}$ for rationale length regularization and $\alpha$ for the weights of RE and KD. We conducted experiments to examine the effects of them on the different variants of neural models (See Appendix D.1 for $\lambda_{\text{select}}$; Appendix D.2 for $\alpha$). More hyperparameter settings can be found in Appendix D.3.

### 3.4. Evaluation Metrics

We discuss rationale extraction evaluation by first criticizing a common belief in the field of XAI: a good rationale should be aligned with human knowledge (Bastings et al., 2019; DeYoung et al., 2020; Yue et al., 2022; Ghasemi Madani & Minervini, 2023) (i.e., often referred to as *plausibility*). In RE, that is to measure the alignment or overlap between machine rationales and human annotated rationales. Instead, we argue that we are supposed to consider rationale extraction as a process of discovering what features are important for what predictions within the training data which is determined by the learning objective of rationale extraction (See the analysis under Equation (5)), and the knowledge in the training data does *not* have to be aligned with human knowledge or annotations. For example, a rationale extraction model (or an XAI method in general) could discover that, given a radiology report, the name of a hospital is leading to a cancer diagnostic prediction (due to some biased training data) while a human would care more about actual symptoms. However, it does not mean the hospital name is bad as a rationale because (1) it truthfully reflects the model's knowledge that is from the training data and (2) this rationale informs humans that this diagnostic prediction may not be reliable. In general, since human knowledge could also to some degree be aligned with the knowledge in the data, the alignment measure could be useful to some degree but should not be viewed as an objective metric.

**An objective evaluation metric**   Given the motivation above, a more objective evaluation metric is to measure the predictive performance on the final task (e.g., classification) given the constraint on the percentage of features that can be utilized as rationales. In this way, RE is viewed as a method for learning interpretable representations of knowledge (i.e., through feature selections) that achieve good predictions

instead of learning knowledge that is aligned with humans'. This objective is more general as it allows neural models to discover new patterns from the data, which could be unknown to humans, and does not require human annotations for evaluation, which are often expensive to acquire.[4] This predictive performance-based evaluation has also been practiced commonly (Bastings et al., 2019; Jain et al., 2020; Dai et al., 2022; Ghasemi Madani & Minervini, 2023). It is also the evaluation metric used in our work.

**Baselines**   Because our work, to the best of our knowledge, is the first approach of knowledge distillation in rationale extraction, we use the students' RE models as the baselines (i.e., we compare REKD to RE).

## 4. Results

In this section, we report the experimental results of REKD, compared to RE, on the three datasets, CIFAR 10/100 and IMDB. The results on pure classification (CLS) are also added as a special case of rationale extraction, i.e., classification is simply having the whole input as the rationale. As previously observed by Jain et al. (2020) and Dai et al. (2022), the predictive performance of rationale extraction is generally positively correlated to the rationale length. We observe this clear pattern when varying the rationale percentage target $p_{\text{target}}$ of ViT RE models on CIFAR 10 (Figure 3). It is reasonable as, by using a small $p_{\text{target}}$, we put a strong constraint on how much information could be utilized for a task prediction, which generally could hurt the predictive performance. As the most important result, REKD significantly improves the predictive performance of the student RE models with reduced variance. See the results reported in Tables 1, 2 and 3. The rationale extraction examples on CIFAR 10, 100 and IMDB can be found in Appendix D.5.

**"Chicken and egg" dilemma vs. model capacity**   One immediate observation is that the predictive performance of RE drops more from classification to a low target rationale ratio when the base neural model is less capable (shown in Figure 3, Table 1, 2 and 3). This phenomenon happens across language and vision domains, with ViTs and BERTs. As an example in Table 1, the predictive accuracy of ViT Small on CIFAR 10 drops significantly from 0.981@CLS to 0.889@RE (i.e., a 0.092 decrease) while the accuracy degeneration for ViT Base is only 0.020 (from 0.984@CLS to 0.964@RE). It aligns with our hypothesis that the "chicken and egg" dilemma is exacerbated when the base neural models are not sufficiently capable. In addition, in Figure 3, the predictive accuracy on CIFAR 10 drops faster for less capable ViT models when we gradually decrease $p_{\text{target}}$ from

---

[4]The Bitter Lesson by Sutton (2019) applies to the discussion on XAI evaluation.

CLS to 5%.

*Table 1.* Predictive performance of ViT CLS, RE and REKD on CIFAR 10. We report accuracy(std) and rationale ratio(std) for each model. The differences of each model's accuracy between CLS and RE and between RE and REKD are indicated with arrows.

| Task | ViT Model | Accuracy | R % |
|------|-----------|----------|-----|
| CLS | Base | .984(.002) | 100% |
| | Small | .981(.001) | 100% |
| | Tiny | .968(.002) | 100% |
| RE | Base (T) | .964(.009); .020 ↓ | 15.2%(.4%) |
| | Small | .889(.019); .092 ↓ | 14.9%(.2%) |
| | Tiny | .797(.054); .171 ↓ | 14.8%(.5%) |
| REKD | Small (S) | .968(.006); .079 ↑ | 14.8%(.2%) |
| | Tiny (S) | .936(.003); .139 ↑ | 14.9%(.3%) |

**CIFAR 10**    Overall, given the same constraint on rationale ratio for REKD and RE, REKD improves ViT Small from 0.889@RE to 0.968 and ViT Tiny from 0.797@RE to 0.936 in accuracy. It is worth mentioning that, by learning from the ViT Base teacher of seed=2029 (with an accuracy of 0.969), the average accuracy of ViT Small with REKD is 0.968, which slightly outperforms the average of the teacher ViT Base models, i.e., 0.964. More results on CIFAR 10 can be found in Table 1.

*Table 2.* Predictive performance of ViT CLS, RE and REKD on CIFAR 100. We report accuracy(std) and rationale ratio(std) for each model. The differences of each model's accuracy between CLS and RE and between RE and REKD are indicated with arrows.

| Task | ViT Model | Accuracy | R % |
|------|-----------|----------|-----|
| CLS | Base | .947(.004) | 100% |
| | Small | .944(.003) | 100% |
| | Tiny | .903(.003) | 100% |
| RE | Base (T) | .830(.028); .117 ↓ | 15.1%(.7%) |
| | Small | .779(.028); .165 ↓ | 14.8%(.9%) |
| | Tiny | .645(.005); .258 ↓ | 14.8%(.5%) |
| REKD | Small (S) | .845(.015); .066 ↑ | 15.2%(.2%) |
| | Tiny (S) | .777(.007); .132 ↑ | 15.1%(.3%) |

**CIFAR 100**    Compared to the results in CIFAR 10 which imposes the same 15% rationale constraint, the overall predictive performance of the ViT RE models degenerates more from CLS to RE. It suggests that, when a RE model is expected to learn a more fine-grained or difficult task (e.g., with 20-class classification in CIFAR 100), a relatively low rationale ratio could be limiting. The ViT Small student achieves a predictive accuracy of 0.845@REKD, compared

to 0.779@RE; ViT Tiny has also been significantly improved in accuracy (0.777@REKD vs. 0.645@RE). Note that the particular ViT Base run (with seed=2031) used as the teacher has an accuracy of 0.848@RE. With the guidance from this particular ViT Base teacher, the average of the ViT Small students even outperforms the average of the ViT Base models (i.e., 0.845 vs. 0.830). Other CIFAR 100 results are reported in Table 2.

*Table 3.* Predictive performance of BERT CLS, RE and REKD on IMDB movie reviews. We report accuracy(std) and rationale ratio(std) for each model. The differences of each model's accuracy between CLS and RE and between RE and REKD are indicated with arrows.

| Task | BERT Model | Accuracy | R % |
|------|-----------|----------|-----|
| CLS | Base | .914(.004) | 100% |
| | Small | .889(.003) | 100% |
| | Mini | .877(.003) | 100% |
| RE | Base (T) | .912(.004); .002 ↓ | 9.7%(.6%) |
| | Small | .881(.005); .008 ↓ | 10.3%(.6%) |
| | Mini | .863(.005); .014 ↓ | 10.2%(.8%) |
| REKD | Small (S) | .906(.002); .025 ↑ | 9.8%(.4%) |
| | Mini (S) | .892(.002); .029 ↑ | 10.0%(.3%) |

**IMDB**    With the constraint of 10% rationale ratio, the BERT models only experience a very slight degeneration in accuracy between CLS and RE: 0.002 for BERT Base, 0.008 for BERT Small and 0.014 for BERT Mini. It suggests that the classification on the IMDB dataset does not require a large number of features for making accurate predictions. Overall, REKD improves BERT Small from 0.881@RE to 0.906 and BERT Mini from 0.863@RE to 0.892. It should be noted that the REKD results are even higher than the student models' CLS results: 0.906 vs. 0.889@CLS for BERT Small and 0.892 vs. 0.877@CLS for BERT Mini. It suggests that the teacher model might deliver rationales with strong predictive patterns that are not acquired by the black box student CLS models. The detailed results are in Table 3.

**Ablation studies**    To better understand each component of REKD, we have conducted ablation studies to answer the following questions: (1) What happens if we perform pure supervised learning, i.e., KD without RE? (2) Does the performance gain come from more stable optimization of supervised learning or the teacher knowledge with predictively powerful rationales? (3) Which component of the knowledge distillation is more important, rationale or prediction distillation? See Appendix C for details.

# 5. Related work

**Rationale extraction**   Rationale extraction with the select-predict architecture was originally proposed by Lei et al. (2016). It can be seen as a mechanism for hard and sparse attention over the input features. The design guarantees that the rationales, i.e., the selected features, are faithful explanations by making predictions solely based on the selected features. However, due to the non-differentiability of the discrete feature selections, Lei et al. (2016) turned to REINFORCE (Williams, 1992) for gradient estimations, which could be problematic for its high variance in training. To ease the training, Jain et al. (2020) proposed to turn RE into a supervised learning task, which obtained rationale-level supervision by selecting features from a trained model using a heuristics-based method, e.g., selecting top-k tokens from a trained BERT based on attention scores (Bahdanau et al., 2016), and then trained the generator and the predictor independently. Their approach is similar to a special case of REKD, i.e., setting $\alpha = 0$. With the development of the Gumbel-Softmax estimator (Jang et al., 2017), later Gumbel-Softmax has become a standard component in RE for differentiable rationale sampling (Paranjape et al., 2020; Xie & Ermon, 2021; Carton et al., 2022; Dai et al., 2022; Jiang et al., 2024; Yue et al., 2024). Our approach implements Straight-Through Gumbel-Softmax (Bengio et al., 2013; Jang et al., 2017), i.e., ensuring that the feature selections are discretized during the training. Regarding the difficulty of training RE, Yu et al. (2021) also discussed the cooperative mechanism between the generator and the predictor, which they referred to as "interlocking", i.e., a predictor can be forced to make predictions based on sub-optimal rationale selections and a generator might be reinforced by that prediction signal; Dai et al. (2022) argued that searching for a good rationale (i.e., proof) is much harder than making a good prediction when provided with a good rationale (i.e., verification). Their insights align with our analysis on the "chicken and egg" dilemma.

**Knowledge distillation**   The original knowledge distillation (KD) was proposed by Hinton et al. (2015). By using a temperature-scaled Softmax, KD obtains the "soft" and informative probability distributions from a teacher model's output logits as guidance for a student's learning in addition to the student's own task objective. Later work on feature-based distillation has shown that distilling the intermediate layers could further improve a student model, e.g., distilling latent feature representations (Romero et al., 2015) and attention maps (Zagoruyko & Komodakis, 2017). However, with the dimension mismatch between a student's and a teacher's latent layers, additional processing is required, such as a learnable projection function (Romero et al., 2015) and channel-collapsing (Zagoruyko & Komodakis, 2017). In rationale extraction, because feature selection serves as a universal interface, REKD has the structural advantage of being robust to architecture mismatch. Jafari et al. (2021) proposed annealing knowledge distillation where temperature annealing helps to bridge the capacity gap between a large teacher and a small student. Though annealing KD is similar to REKD in terms of the soft-to-hard distribution transition, their scheduler serves as a heuristic curriculum for capacity-limited students; conversely, REKD imposes annealing as a structural requirement to enforce distributional consistency with the student's Gumbel-Softmax trajectory, inherently generating a curriculum as a byproduct.

# 6. Limitations and Discussion

**Rationale quality**   As discussed in Section 3.4, human annotation alignment metrics can unfairly penalize a faithful model for exposing genuine dataset biases, and evaluating predictive performance under a rationale constraint provides a more objective measure of rationale quality. Confirming rationale quality on real-world datasets is inherently difficult without ground-truth feature maps, and the cooperative nature of the select-predict architecture remains vulnerable to learning covert communication channels instead of extracting informative structures (Wäldchen et al., 2024; Liu et al., 2025). We recognize that while human annotations are flawed as an objective ground truth, human-centric usability studies are a valuable step for the real-world adaptation of RE. Complementary to this, once a high-capacity RE teacher is validated, REKD utilizes those teacher rationales for student supervision. The knowledge distillation in REKD acts as a strong regularization on the student's feature selection, restricting its ability to form arbitrary communication channels and yielding improved rationale quality.

**Distilling across architectures**   REKD is neural-model agnostic by design because the intermediate feature selection layer serves as a universal interface for the teacher and student neural networks. However, in our experiments, distillation is only performed between two neural networks of the same general architecture, e.g., two ViTs or two BERTs, even though they have different hidden dimensions and numbers of layers. Further work could explore if the method generalizes to distillation between neural architectures with fundamentally different inductive biases, e.g., a ViT and a ResNet. Crucially, this requires that the different neural architectures actually share the same input features, e.g., sharing identical tokenization processes for language tasks.

**Potential applications**   The direct application of REKD could be in scenarios where interpretability is important, but the computational resources are limited, such as mobile devices for healthcare (Hernández-Neuta et al., 2019). As a possible direction for future work, our method could also be applied to distilling discrete latent structures that are learned

with Gumbel-Softmax, e.g., relational graphs (Kipf et al., 2018).

## 7. Conclusion

This paper proposes Rationale Extraction with Knowledge Distillation (REKD) which is a neural-model agnostic distillation approach for Gumbel-Softmax-based rationale extraction. The motivation aligns well with how effectively humans could learn from a teacher who provides interpretable and verifiable knowledge that helps to break the "chicken and egg" dilemma. By synchronizing the knowledge distillation temperature with the Gumbel-Softmax annealing, it allows the students to learn with a KD curriculum in addition to its own RE exploration. The experiments across language (IMDB) and vision (CIFAR 10/100) domains with various BERT and ViT models show that the approach significantly improves the predictive performance of lightweight student RE models.

## Acknowledgements

Alberta Machine Intelligence Institute (Amii) provided the compute resources, and this research was supported by both Amii and the Natural Sciences and Engineering Council of Canada (NSERC).

## Impact Statement

This paper presents work whose goal is to advance the field of interpretable neural networks and machine learning. There are many potential societal consequences of our work, none of which we feel must be specifically highlighted here.

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

## A. Base Model and Dataset Accessibility

All the base neural models are publicly available through HuggingFace with the following model labels "bert-base-uncased", "prajjwal1/bert-small", "prajjwal1/bert-mini", "google/vit-base-patch16-224", "WinKawaks/vit-small-patch16-224", "WinKawaks/vit-tiny-patch16-224".

All the datasets are also publicly available through HuggingFace with the following dataset labels "imdb", "cifar10" and "cifar100".

## B. Gradient Magnitude for Distilling Gumbel-Softmax Distributions

We first show that the gradient magnitude of the Gumbel-Softmax estimator scales inversely with temperature ($\propto 1/\tau$). Let $m_i^{(S)}$ denote the sum of student logit $z_i^{(S)}$ and its corresponding Gumbel noise and $m_i^{(T)}$ for the sum of the teacher logit and its Gumbel noise. Then, we get the probabilities $q_i^{(S)}$ and $q_i^{(T)}$ by applying a $\tau$-scaled softmax:

$$q_i^{(S)} = \frac{e^{m_i^{(S)}/\tau}}{\sum_j e^{m_j^{(S)}/\tau}}, \quad q_i^{(T)} = \frac{e^{m_i^{(T)}/\tau}}{\sum_j e^{m_j^{(T)}/\tau}}$$

By following the proof by Hinton et al. (2015), given the distillation loss $\mathcal{L}$ (i.e., either cross-entropy or KL divergence), we have

$$\frac{\partial \mathcal{L}}{\partial m_i^{(S)}} = \frac{1}{\tau}(q_i^{(S)} - q_i^{(T)})$$

As $m_i^{(S)}$ and $z_i^{(S)}$ are different by a constant Gumbel noise (i.e., $\frac{\partial m_i^{(S)}}{\partial z_i^{(S)}} = 1$), we also have

$$\frac{\partial \mathcal{L}}{\partial z_i^{(S)}} = \frac{\partial \mathcal{L}}{\partial m_i^{(S)}} \frac{\partial m_i^{(S)}}{\partial z_i^{(S)}} = \underbrace{\frac{1}{\tau}}_{\text{Scaling Factor}} \cdot \underbrace{(q_i^{(S)} - q_i^{(T)})}_{\text{Standard Softmax Gradient}}$$

which is the result of the gradient of a standard Softmax being scaled by $\frac{1}{\tau}$. In the original proof (Hinton et al., 2015), when $\tau$ is a fixed high value, the scale is near 0, which would cause vanishing gradient for the distillation loss, compared to the scale of the standard Softmax gradient for the task loss. Hence, a $\tau^2$ scale was added to the distillation loss term to balance the gradient magnitudes. See more detailed derivation in the original proof with the high $\tau$ and zero-mean logits assumptions.

However, for rationale extraction, because the task loss, i.e., $\mathcal{L}_{\text{RE}}$ in Equation (5), has the gradients (w.r.t to the generator network parameters) also scaled by $\frac{1}{\tau}$ due to the use of Gumbel-Softmax, the rationale distillation gradients' magnitude has been balanced with the task gradients'. If we added a scale of $\tau^2$ to the rationale distillation loss, we would have

$$\frac{\partial \mathcal{L}_{\text{new}}}{\partial z_i^{(S)}} = \tau^2 \frac{\partial \mathcal{L}}{\partial z_i^{(S)}} = \tau(q_i^{(S)} - q_i^{(T)}) \to 0$$

as $\tau \to 0$ during the annealing process, which would cause gradient vanishing. Instead, we use a hyperparameter $\lambda_R$ and rely on Gumbel-Softmax itself to handle the gradients internally.

## C. Ablation Studies

We investigate the following variants of REKD to better understand the contribution of each component of our design. Except the component to be tested, all settings are the same as in REKD.

**KD without RE**    Given a teacher RE model, can we simply turn the RE learning of student into a pure supervised learning problem? It could also help to break the "chicken and egg" dilemma. By setting $\alpha = 0$ in REKD loss function of Equation (10), REKD becomes a completely supervised learning task. That is the generator and the predictor are trained independently with their separate supervision signals from the teacher. However, experiments show that it is generally not as good as using other choices of $\alpha$ of REKD (see Table 4 for accuracy, Tables 9 and 10 for dev loss). It indicates that a student's own exploration is also beneficial.

*Table 4.* Comparing the predictive accuracy of REKD and pure supervised learning (KD only by setting $\alpha = 0$) on CIFAR 10.

| Variant | REKD | pure supervised |
|---|---|---|
| ViT Small | **.968@14.8** | .967@14.8 |
| ViT Tiny | **.936@14.9** | .928@14.8 |

**Stable optimization vs. teacher knowledge**  Does the performance gain of REKD come from (1) teacher rationales with high predictive power or (2) more stable optimization from the supervised learning? We designed an experiment to roughly answer this question where we make a student model learn from its own RE (i.e., supervision provided but not knowledge from a powerful teacher). We observed that the performance ordering is REKD(teacher_distillation) $\gg$ self_distillation $>$ RE for both ViT Small and Tiny on CIFAR10. It shows that both (1) and (2) contribute, but (1) is more influential. See Table 5 for the details.

*Table 5.* Comparing the predictive accuracy of self-distillation, REKD and RE with ViT Small and Tiny on CIFAR 10. Each experiment was performed 10 times for an averaged value. The best performing entry is in bold for each base network.

| Variant | REKD | self-distillation | RE |
|---|---|---|---|
| ViT Small | **.968@14.8** | .911@15.1 | .889@14.9 |
| ViT Tiny | **.936@14.9** | .82@15 | .797@14.8 |

**Separate distillation**  Which component of the knowledge distillation is essential, rationale or prediction distillation? Intuitively, rationale distillation is more important as it brings the predictively powerful rationales from a teacher and helps to break the "chicken-and-egg" dilemma. The experimental results on distilling rationales and predictions separately using ViT Small and Tiny on CIFAR 10 verify our intuition. See Table 6.

*Table 6.* Comparing the predictive accuracy of REKD, rationale-only distillation and prediction-only distillation with ViT Small and Tiny on CIFAR 10. Each experiment was performed 10 times for an averaged value. The best performing entry is in bold for each base network.

| Variant | REKD | rationale-only | prediction-only |
|---|---|---|---|
| ViT Small | **.968@14.8** | .967@14.9 | .702@14.2 |
| ViT Tiny | **.936@14.9** | .930@15 | .65@14.8 |

## D. More Experimental Details

All experimental results were averaged over 10 runs, if not otherwise specified, with random seeds $= [2026, 2027, ..., 2035]$. We used up to 120 pieces of L40S 48 GB simultaneously for running different tasks and seeds in parallel. One single L40S with 48 GB memory was sufficient for each experiment.

### D.1. Tuning $\lambda_{\text{select}}$

The $\lambda_{\text{select}}$ that achieves the lowest RE loss on the developing data is chosen as long as the rationale ratio does not get far from $p_{\text{target}} = 15\%$ for CIFAR or $p_{\text{target}} = 10\%$ for IMDB ($> 1\%$). The detailed results can be found in the Table 7 and 8. The entry in bold for each model corresponds to the final $\lambda_{\text{select}}$ choice. The $\lambda_{\text{select}}$ decided in CIFAR 10 is then also used for CIFAR 100.

### D.2. Tuning $\alpha$

The $\alpha$ (i.e., the scale for $\mathcal{L}_{\text{RE}}$) that achieves the lowest RE loss on the developing data is chosen. The detailed results can be found in the Table 9 and 10. The entry in bold (i.e., the lowest develop RE loss) for each model corresponds to the final $\alpha$ choice (i.e., $\alpha = 0.3$ is then used for all models, including BERTs).

### D.3. Other Hyperparameters

AdamW is used as the optimizer for all models with a weight decay rate $= 0$ for BERTs and 1e-3 for ViTs. The dropout rates for BERTs and ViTs are separately 0.5 and 0.1.

## D.4. RE Rationale Ratio vs. Accuracy

The details of rationale ratio vs. accuracy for ViT models in CIFAR 10 for Figure 3 are reported in Table 11.

*Table 7.* Tuning $\lambda_{\text{select}}$ for ViT RE with $p_{\text{target}} = 15\%$: rationale extraction loss of ViT models on the developing data for CIFAR 10. Each entry is of format (rationale_ratio; dev_loss). The lowest dev loss with rationale ratio not far from target (within 1%) is in bold.

| $\lambda_{\text{select}}$ | 1e-4 | 5e-4 | 1e-3 | 5e-3 | 1e-2 | 5e-2 |
|---|---|---|---|---|---|---|
| ViT Base | 18.5%; 0.2386 | 16%; 0.1745 | 15.2%; 0.1727 | **15.2%; 0.1462** | 15%; 0.1498 | 15%; 0.2747 |
| ViT Small | 14.8%; 0.742 | 14.1%; 0.5163 | 14.4%; 0.4427 | 14.7%; 0.4109 | **14.9%; 0.3444** | 14.8%; 0.369 |
| ViT Tiny | 15.6%; 0.834 | 13.2%; 0.788 | 13.5%; 0.7082 | 14%; 0.7364 | 13.8%; 0.7996 | **14.8%; 0.6994** |

*Table 8.* Tuning $\lambda_{\text{select}}$ for BERT RE with $p_{\text{target}} = 10\%$: rationale extraction loss of BERT models on the developing data for IMDB movie reviews. Each entry is of format (rationale_ratio; dev_loss). The lowest dev loss with rationale ratio not far from target (within 1%) is in bold.

| $\lambda_{\text{select}}$ | 1e-5 | 5e-5 | 1e-4 | 5e-4 | 1e-3 | 5e-3 |
|---|---|---|---|---|---|---|
| BERT Base | 11.2%; 0.2515(9 runs) | **9.7%; 0.2524** | 9%; 0.2579 | 9.2%; 0.2636 | 9.2%; 0.2773 | 9%; 0.3114 |
| BERT Small | 13.2%; 0.3104(8 runs) | 11.3%; 0.3104(7 runs) | **10.7%; 0.3149** | 9.3%; 0.3298 | 9.3%; 0.3363 | 10.1%; 0.3537 |
| BERT Mini | 12%; 0.3456 | 11.1%; 0.3479 | **10.3%; 0.347** | 9.7%; 0.3538 | 9.7%; 0.3632 | 9.6%; 0.4019 |

*Table 9.* Tuning $\alpha$ for ViT REKD in CIFAR 10: rationale extraction loss of ViT models on the developing data.

| $\alpha$ | 0 | 0.3 | 0.5 | 0.7 |
|---|---|---|---|---|
| ViT Small | 0.159 | **0.145** | 0.146 | 0.151 |
| ViT Tiny | 0.315 | **0.271** | 0.277 | 0.280 |

*Table 10.* Tuning $\alpha$ for ViT REKD in CIFAR 100: rationale extraction loss of ViT models on the developing data.

| $\alpha$ | 0 | 0.3 | 0.5 | 0.7 |
|---|---|---|---|---|
| ViT Small | 0.634 | **0.563** | 0.568 | 0.568 |
| ViT Tiny | 0.929 | **0.784** | 0.787 | 0.795 |

*Table 11.* ViT rationale percentage vs. accuracy for CIFAR 10 classification by varying $p_{\text{target}}$ from 5% to 75%. Each entry is of format: rationale percentage;accuracy.

| $p_{\text{target}}$ | 5% | 15% | 25% | 35% | 45% | 55% | 65% | 75% | cls |
|---|---|---|---|---|---|---|---|---|---|
| ViT Base | 5.6;0.915 | 15.2;0.964 | 25.1;0.961 | 34.9;0.963 | 45.0;0.967 | 55.1;0.972 | 65.2;0.977 | 75.3;0.98 | 100;0.984 |
| ViT Small | 4.7;0.703 | 14.9;0.889 | 24.8;0.933 | 34.9;0.954 | 45.0;0.966 | 55.1;0.97 | 65.4;0.972 | 75.2;0.977 | 100;0.981 |
| ViT Tiny | 4.8;0.634 | 14.8;0.797 | 24.8;0.879 | 35.0;0.908 | 45.0;0.935 | 55.0;0.95 | 65.2;0.953 | 75.3;0.96 | 100;0.968 |

## D.5. RE Examples

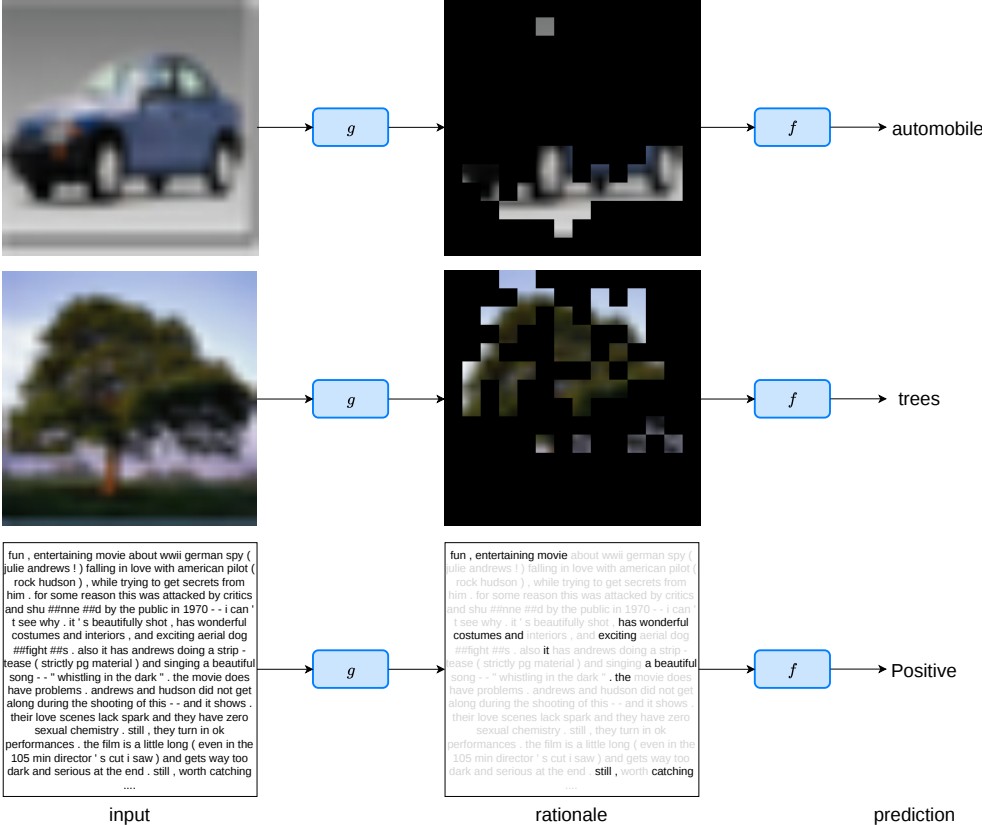

*Figure 4.* Examples of rationale extraction for ViT Base on CIFAR 10, CIFAR 100 and BERT Base on IMDB movie reviews (arranged from top to bottom).

