# OpenReview forum: "Learn from A Rationalist: Distilling Intermediate Interpretable Rationales"
_ICML.cc/2026/Conference — ICML 2026 regular_

### Official Review · Reviewer_K5Lf · 2026-03-02

**Soundness:** 3
**Presentation:** 3
**Significance:** 3
**Originality:** 3
**Overall Recommendation:** 4
**Confidence:** 3

**Summary:**

The paper addresses the "chicken and egg" optimization challenge in rationale extraction (RE), where a generator and predictor must learn jointly via remote supervision—a task that is particularly difficult for less capable or smaller neural networks. To resolve this, the authors propose Rationale Extraction with Knowledge Distillation (REKD), a framework allowing a student RE model to learn feature selections and predictions from a trained teacher model alongside its own exploration. The authors evaluate REKD across language (IMDB) and vision (CIFAR 10/100) classification tasks using BERT and ViT variants. The results indicate that REKD significantly enhances the predictive accuracy of lightweight student RE models compared to standard RE baselines.

**Compliance With Llm Reviewing Policy:**

Affirmed.

**Key Questions For Authors:**

1. The experiments currently perform distillation between identical architecture families (e.g., ViT Base to ViT Small). Have you conducted any preliminary experiments on cross-architecture distillation (e.g., ResNet to ViT) to fully validate the "neural-model agnostic" claim?
2. The method uses fixed weights ($\alpha$) for the RE and KD loss terms. Did you experiment with dynamic weight scheduling (e.g., prioritizing KD early on and autonomous RE later), and if so, how did it impact convergence?
3. While you argue against the necessity of human alignment for rationales, how does REKD perform on standard plausibility metrics (e.g., the ERASER benchmark) compared to your RE baseline? Does teacher guidance inadvertently improve human alignment?

**Strengths And Weaknesses:**

Strengths:
1. The synchronization of the knowledge distillation temperature with the Gumbel-Softmax annealing scheduler creates a natural and elegant curriculum learning strategy. This mechanism effectively allows the student model to transition smoothly from absorbing broad, softened teacher probabilities to mimicking sharp, discrete feature selections as the temperature decays.
2. In specific configurations, the student models guided by a strong teacher manage to slightly outperform the average predictive performance of the teacher models themselves, underscoring the method's efficacy.
3. Because the rationale extraction feature selection layer acts as a universal interface, the framework permits knowledge transfer regardless of the underlying neural architectures of the teacher and student. This design elegantly bypasses the dimensional mismatch issues and the need for complex projection functions that frequently complicate standard intermediate latent layer distillation.

Weaknesses:
1. The authors explicitly dismiss "plausibility" (alignment with human annotations) as an objective metric, relying entirely on final task predictive performance under a rationale length constraint. By ignoring human alignment, the methodology fails to verify if the distilled rationales are actually interpretable, faithful, or useful to human end-users, which contradicts the primary motivation for employing XAI in high-stakes domains.
2. The distillation experiments are strictly homogeneous (e.g., BERT-to-BERT, ViT-to-ViT), leaving the claim that the framework is fully neural-model agnostic untested in practical, cross-architecture scenarios. The evaluated datasets (CIFAR 10/100 and IMDB) are balanced, standard benchmarks; it remains unclear how the method performs on heavily skewed, complex, or high-stakes datasets mentioned in the introduction.
3. The rationale distillation weight $\lambda_R$ is fixed at a constant 0.5 across all experiments without an accompanying ablation study to justify this specific value.

---

> ### Author Rebuttal · Authors · 2026-03-31
>
> We thank the reviewer for their valuable feedback.
>
> **Weakness responses:**
>
> 1. Interpretability(a)/faithfulness(b)/usefulness(c):
>
>    (a) We respectfully clarify that not aligning with human rationale annotations does not mean not interpretable by humans and RE is regarded as self-interpretable due to the select-predict architecture. Consider that there is a hospital famous for treating cancer and almost all patients from that hospital are positive cancer cases (as the diagnosed patients go there to get cancer treated well). When the radiology report data from this hospital is included in a dataset to learn a statistical model (e.g., a NN) for cancer diagnostic predictions, the predictive model could likely learn that the prediction is cancer whenever that hospital’s name appears. That is a RE model would select the hospital’s name as a rationale and make a cancer prediction. It is interpretable as RE tells that the hospital name is the reason for the prediction. In this case, while the model’s rationale is different from a human physician’s rationale (i.e., probably on medical symptoms), the physician could still make sense of the rationale and the rationale serves as a diagnostic tool for the physician to decide if the diagnostic prediction is trustworthy or not. We have provided the related discussion in Section 3.4 to demonstrate this point from a knowledge acquisition perspective.
>
>    (b) The faithfulness of a rationale is guaranteed in RE because the selected features in a rationale are truly the important features for a prediction (and not selected features simply have no effect on the prediction). Please also see related discussion in the Weakness 2 response to reviewer hZs6.
>
>    (c) The usefulness of a rationale in RE is that it explicitly tells humans what features are used for a prediction, which offers a diagnostic tool for model predictions. Further comprehensive human confirmation over the REKD vs. RE rationales to support the value of the improved rationales is interesting but is likely too much for the timetable for this rebuttal. We will acknowledge it in paper future work.
>
> 2. (a) Model-agnostic: See Question 4 response to Reviewer iSGE
>
>    (b) More complex dataset: Our paper is to provide the evidence for the general viability of REKD through experiments with multiple popular variants of BERTs and Vision Transformers on popular language and vision tasks. We acknowledge that experimenting with more complex datasets and fundamentally different neural families could further enhance the viability of REKD. However, it is likely out of the scope of our paper and will be acknowledged in future work.
>
> 3. Justify $\lambda_R$: We actually did some preliminary experiments on the effect of $\lambda_R$ during the REKD development. See the results on CIFAR 10 developing loss of ViT Small and Tiny. Among three experimented options (0.05, 0.1, 0.5), $\lambda_R$=0.5 achieved the best developing loss for both ViT Small and Tiny on CIFAR10. We believe more $\lambda_R$ tuning would only further enhance REKD performance. See results below (averaged over 10 runs).
>
>    For ViT Small (Dev loss): $\lambda_R$=0.05 is .254; $\lambda_R$=0.1 is .201; $\lambda_R$=0.5 is .155
>
>    For ViT Tiny: $\lambda_R$=0.05 is .578; $\lambda_R$=0.1 is .514; $\lambda_R$=0.5 is .39
>
> **Question responses:**
>
> 1. Model-agnostic: See Question 4 response to Reviewer iSGE
>
> 2. RE/KD dynamic weights: We acknowledge that experimenting with different weight schedulers for RE and KD loss terms is interesting, such as more KD in an early stage and more RE in a later stage, because it aligns with the intuition that supervised learning first + exploration after could enhance the generalization of reasoning models (e.g., the recently emerging large `reasoning' models often apply a two-stage training strategy: pretraining + reinforcement learning). In the future work section, we have also pointed out that a dynamic weighting strategy could possibly further enhance REKD, which is empirical and deserves future investigation.
>
> 3. ERASER alignment: We thank the reviewer for understanding our argument against chasing alignment with human rationale annotations. We acknowledge that the ERASER datasets could be used to check if REKD rationales show improved human rationale alignments inadvertently. However, it is likely out of the scope of our paper and might be infeasible within the limited rebuttal time. We will acknowledge it as future work.

---

> > ### Author Rebuttal · Reviewer_K5Lf · 2026-04-02
> >
> > I appreciate your thoughtful and detailed replies to my questions and comments. You have addressed all of my concerns satisfactorily, and I have no further queries.

---

### Official Review · Reviewer_BzfV · 2026-03-12

**Soundness:** 3
**Presentation:** 3
**Significance:** 2
**Originality:** 2
**Overall Recommendation:** 3
**Confidence:** 3

**Summary:**

This paper studies rationale extraction models with a select-predict architecture, focusing on the practical difficulty of training the generator and predictor jointly when the backbone model is small. The authors propose REKD, where a student rationale-extraction model is trained with its standard rationale-extraction objective while also distilling two signals from a stronger teacher rationale-extraction model: the teacher’s feature-selection distributions and the teacher’s prediction distribution. A central design choice is to tie the distillation temperature to the Gumbel-Softmax temperature annealing used for discrete feature selection, with the goal of providing softer guidance early and sharper guidance later. Experiments on IMDB and CIFAR-10/100 with BERT and ViT teachers and smaller student variants report substantial accuracy improvements at fixed rationale budgets, and reduced variance across random seeds.

**Compliance With Llm Reviewing Policy:**

Affirmed.

**Final Justification:**

Thanks for the rebuttal. I appreciate the authors' rebuttal.

I decide to keep my current rating.

**Key Questions For Authors:**

1. How is the teacher model chosen for distillation in each setting, and how sensitive are the student results to that choice? If you rerun REKD with multiple teacher seeds or multiple teacher checkpoints, do the gains remain consistent, and what is the variance attributable to the teacher?
2. Please clarify the exact train, development, and test splits for IMDB, and confirm whether evaluation is performed on the standard held-out test set or on a subset split from training.
3. Can you provide ablations that isolate the contributions of each part of REKD: prediction distillation only, feature-selection distillation only, and the effect of tying the distillation temperature to the Gumbel-Softmax annealing versus using a separate schedule?
4. How does REKD compare against stronger training baselines for rationale extraction beyond vanilla joint training, such as pseudo-rationale supervision approaches or other established strategies to reduce training variance? A broader comparison is important to validate the claimed advantage.
5. Beyond task accuracy under a rationale budget, what evidence supports that the learned rationales are better as explanations? For example, do you observe improved stability across runs, more coherent selections, or stronger faithfulness-style tests such as comprehensiveness or sufficiency analyses?

**Limitations:**

yes

**Strengths And Weaknesses:**

Strengths

- The approach is technically straightforward and, as described, appears sound. The objective is clearly defined, and distilling both intermediate feature-selection behavior and final predictions is a reasonable way to stabilize and strengthen learning in a select-predict setup. The temperature-coupling idea is also intuitive given the role of annealing in Gumbel-Softmax training.

- Empirically, the gains are large, especially for the smaller students where standard rationale extraction degrades sharply relative to full-input classification. Reporting results over 10 seeds and showing variance reductions strengthens the claim that teacher guidance improves stability, not only mean performance. Demonstrating the method in both language and vision settings is also a plus.

Weaknesses

- The evaluation and comparisons are too narrow for the strength of the claims. The main baseline is the same student trained with rationale extraction only. It is hard to attribute the improvements to the proposed design choices without stronger ablations, such as distilling only predictions, distilling only feature selections, or decoupling the two temperatures. The paper argues that task accuracy under a rationale budget is the key objective metric, but it does not provide much evidence in the main text about the resulting rationales as explanations, for example their stability, coherence, or usefulness to humans in practice.

- There are also experimental-design choices that need clearer justification. Using CIFAR-100 with 20 coarse classes is nonstandard and makes it harder to compare to related work. The IMDB split description is ambiguous, and the procedure for selecting the teacher checkpoint or seed used for distillation should be stated clearly to rule out unintentional cherry-picking. Finally, the “model-agnostic” claim is only partially supported by experiments that stay within the same architecture family; this is a reasonable first step, but it should be framed more carefully.

- The paper has a solid and simple idea with strong-looking results, but it needs broader comparisons, clearer experimental reporting, and a more complete evaluation of the rationales to fully justify its impact.

---

> ### Author Rebuttal · Authors · 2026-03-31
>
> We thank the reviewers for the valuable feedback.
>
> **Weakness responses:**
>
> 1. (a) Separate distillation: We agree that the suggested experiments help to understand the contribution of each component. We conducted the following experiments on CIFAR10: distilling only rationales and only predictions. The results are reported below (10 run average).
>
>    For ViT Small (Acc@R%): REKD is .968@14.8; Distil Rationale is .967@14.9; Distil Prediction is .702@14.2.
>
>    For ViT Tiny: REKD is .936@14.9; Distil Rationale is .930@15; Distil Prediction is .65@14.8.
>
>    The results confirm that rationale distillation brings most of the gain in REKD. It is intuitive as rationale supervision brings knowledge on predictively powerful rationales and breaks the chicken-and-egg dilemma. Also, distilling both rationales and predictions (i.e., original REKD) is slightly better than only distilling rationales.
>
>    (b) Temperature decoupling: using decoupled temperatures for Gumbel-Softmax and KD would introduce more hyperparameters (and more tuning) and is not as simple and elegant as a unified temperature. We agree that using separate temperature settings for KD and Gumbel-Softmax is worth exploring as it also provides the student with supervision and helps to understand the curriculum's effect, but the double temperature scheduler tuning might make it infeasible given the limited rebuttal time. We acknowledge it as future work.
>
>    (c) Rationale quality: the usefulness of a rationale is that it tells what features are used for what predictions, which is guaranteed by the design of RE. Further comprehensive human confirmation over the REKD vs. RE rationales to support the value of the improved rationales is interesting but is likely too much for the timetable for this rebuttal. We will acknowledge it in paper future work.
>
> 2. (a) 20-class, IMDB: When experimenting with 100 classes for CIFAR100, we observed that ViT Small slightly outperforms ViT Base on CLS and RE, which made knowledge distillation experiments meaningless in this setting.
>
>    We apologize for a typo in the IMDB split description: the 20% from the original training data was actually used as dev data (not testing data) and the testing data was the original test split of IMDB.
>
>    (b) Teacher seed: See Weakness 2 response to Reviewer iSGE where experiments show REKD is robust to teacher seed choice.
>
>    (c) Model-agnostic: See Question 4 response to Reviewer iSGE
>
> 3. Thank you for acknowledging the idea of the paper. We agree that the paper presentation would benefit from more experiments and clearer description. We have added the following suggested experiments in the rebuttal: (1) only distilling rationales, (2) only distilling predictions and (3) distilling from different teacher seeds, which will be reflected in the paper. We will also incorporate the rebuttal discussion, regarding datasets and rationale quality, in the paper main text.
>
> **Question responses:**
>
> 1. Teacher seed: See Weakness 2(b) response
>
> 2. IMDB: See Weakness 2(a) response
>
> 3. Separate distillation, decoupled temperature: See Weakness 1(a), 1(b) responses
>
> 4. Pseudo-rationale supervision: To the best of our knowledge, our work is the first on augmenting RE with KD. which limits the comparisons.  In work [1], by obtaining features with top-K attention scores, a pure supervised learning method for RE was performed. It is very similar to a special case of REKD where the RE loss weight alpha = 0 (see discussion in paragraph ``KD without RE” and Related Work in the paper). We have conducted experiments to show that pure supervised learning (alpha=0) is not as good as REKD (alpha=0.3). It could be attributed to that REKD allows the student’s own exploration in addition to KD. The experimental results for ViT Small and Tiny on CIFAR10 are reported (averaged with 10 runs).
> For ViT Small (Acc@R%): pure supervised is .967@14.8; REKD is .968@14.8.
>
> For ViT Tiny: pure supervised is .928@14.8; REKD is .936@14.9.
>
> 5. See Weakness 1(c) response regarding rationale quality. Regarding faithfulness, we respectfully clarify that RE models fundamentally differ from post-hoc feature-importance methods. Because RE architecture follows a select-predict pipeline, the faithfulness of the rationale is structurally guaranteed: the predictor only has access to the selected features, meaning unselected features have zero effect on the predictor’s behavior. Therefore, unlike post-hoc explanations (such as SHAP or LIME) which require rigorous sufficiency testing to show they align with a black-box model's inner workings, RE’s rationales are exactly what the predictor uses.
>
> [1] Jain, S., Wiegreffe, S.,Pinter, Y., and Wallace, B.C (2020). Learning to faithfully rationalize by construction. URL: https://aclanthology.org/2020.acl-main.409.pdf

---

> > ### Author Rebuttal · Reviewer_BzfV · 2026-04-04
> >
> > Thank you for the detailed response, and some of my concerns have been addressed. I will maintain my initial rating.

---

> > > ### Author Response · Authors · 2026-04-07
> > >
> > > Thank you for the reply. Specific points/questions regarding your unsolved concerns would be greatly appreciated.

---

### Official Review · Reviewer_iSGE · 2026-03-12

**Soundness:** 3
**Presentation:** 3
**Significance:** 3
**Originality:** 3
**Overall Recommendation:** 4
**Confidence:** 4

**Summary:**

This paper proposes REKD (Rationale Extraction with Knowledge Distillation), a framework for improving the predictive performance of lightweight rationale extraction (RE) models by distilling knowledge from a stronger teacher RE model. The core observation is that the standard select-predict training objective suffers from a "chicken and egg" problem — the generator needs the predictor's signal to identify important features, but the predictor can only learn well if the generator already selects good features — and that this problem is particularly severe for smaller, less capable networks. The proposed solution has two components: distilling the teacher's Gumbel-Softmax feature selection distributions to the student generator (via KL divergence), and distilling the teacher's prediction distributions to the student predictor (standard KD). The technically interesting aspect is that both RE training and KD naturally share a temperature parameter τ, allowing the authors to synchronize them through a single annealing schedule that creates a curriculum learning effect as a structural byproduct. Experiments are conducted on IMDB (BERT variants) and CIFAR 10/100 (ViT variants), showing substantial accuracy improvements for student RE models under fixed sparsity constraints.

**Compliance With Llm Reviewing Policy:**

Affirmed.

**Key Questions For Authors:**

1. The teacher model is drawn from a single seed for each task and used across all student runs. How sensitive are the student results to the choice of teacher seed? A teacher that happens to have learned cleaner rationales could provide systematically better guidance. Reporting results across multiple teacher seeds, or using an ensemble teacher, would clarify how robust the method is to this choice.

2. The paper distills generator and predictor jointly. Is there an ablation where only generator distillation is applied (with student predictor learning from its own RE loss), or vice versa? Understanding the relative contribution of each distillation component would clarify whether the gain is primarily driven by the feature selection guidance or the prediction guidance.

3. The evaluation metric argument in Section 3.4 is reasonable, but prediction accuracy under a sparsity constraint measures how well the selected features predict, not whether the selected features are stable, diverse, or coherent. Have the authors inspected the rationale visualizations... to verify that REKD students are learning meaningfully different feature selections — I'm not sure the current accuracy numbers alone can confirm this?

4. The paper is restricted to same-architecture teacher-student pairs (ViT-to-ViT, BERT-to-BERT). The conclusion mentions cross-architecture distillation (ViT-to-ResNet) as future work. Is there a principled reason why this should or should not work, given that the feature selection mask is the interface? Some theoretical discussion here would help scope the claim of architecture-agnosticism more precisely.

5. BERT-Mini with REKD achieves 0.892 on IMDB, which exceeds the BERT-Mini CLS baseline of 0.877. This is a surprising result — a model using 10% of tokens outperforms the same model using all tokens. Do the authors have an explanation for this? I'm not sure if this is a regularization effect, a feature of the IMDB dataset specifically, or something about how the masked embeddings interact with BERT's attention mechanism?

6. The experiments are limited to binary sentiment classification (IMDB) for language and CIFAR 10/100 for vision — all relatively simple benchmarks. Do the results hold on more challenging NLP benchmarks with more classes or longer documents (e.g. multi-class classification, legal, or biomedical text)? Similarly, for vision, does the method scale to ImageNet-level class counts and dataset sizes, given that the teacher's rationale quality — on which the student depends entirely — may itself degrade on harder tasks, which is precisely the setting where the guidance would be most needed?

**Limitations:**

The authors do not include a limitations section, which is a real omission. The most significant gap is the restriction to classification tasks and relatively small, well-studied benchmarks. The claim that the method is "neural-model agnostic" is supported in principle but only validated within architecture families, not across them. The evaluation metric argument — while interesting — is not stress-tested against any domain where plausibility vs. faithfulness actually diverges in practice. These are not fatal weaknesses, but they should be acknowledged directly rather than deferred to future work.

**Strengths And Weaknesses:**

Strengths:

1. The chicken-and-egg framing is not new, but tying it specifically to model capacity is cleaner than what prior work did, and the Newton analogy actually works as an intuition.

2. The synchronized temperature annealing is the strongest technical contribution. The point that the τ² correction from standard KD would cause gradient vanishing here is subtle and correctly argued in Appendix B. The contrast with Jafari et al. (2021) is well-drawn.

3. Distilling through the binary mask means the method works across different backbone sizes without needing extra projection layers — unlike FitNets or attention transfer.

4. Results are strong. ViT-Tiny going from 0.797 to 0.936 on CIFAR-10 is a big jump. The BERT-Mini result — where the model using only 10% of tokens actually beats the full-input baseline — is the most interesting finding and deserves more attention than it gets.

5. Reproducibility is handled well. Exact model names, random seeds, hardware, and full hyperparameter sweeps are all provided.

Weaknesses:

1. The claim that the chicken-and-egg problem gets worse with smaller models is only shown through accuracy patterns, not directly tested. I think training instability or optimization differences could explain the same observations.

2. A single teacher seed is used for all student runs with no justification. The CIFAR-100 result where the student beats the average teacher is hard to interpret without knowing how much teacher quality varies across seeds.

3. Generator and predictor distillation are always applied together. Since the generator part is the novel contribution, testing each one separately would make the results more convincing.

4. With 10 runs already done, significance testing on the BERT results would be easy to add and is missing — it seems like the margins there are narrow enough that it matters.

5. The argument against using human-annotated rationales as an evaluation metric is interesting but only gets one paragraph and the paper never actually tests in a setting where this distinction is meaningful.

6. There is no discussion of the cost of training a large RE teacher versus just training a bigger student model directly, which is the obvious practical alternative.

7. All vision experiments are on CIFAR-10 and CIFAR-100, which are small and relatively easy datasets. There are no experiments at ImageNet scale, which is the standard benchmark for vision and the setting where deploying lightweight interpretable models actually matters. Similarly, the largest model used is BERT-base at around 110M parameters and ViT-base at around 86M — both modest by current standards. It is unclear whether the method holds up with larger modern architectures, or whether the teacher's rationale quality remains reliable as model and dataset complexity scale up substantially.

---

> ### Author Rebuttal · Authors · 2026-03-31
>
> We thank the reviewer for the valuable feedback. We are glad that you liked the Issac Newton analogy :)
>
> **Weakness responses:**
> 1. Better optimization?: We suppose both (1) teacher rationales with high predictive power and (2) easier optimization from the supervised learning contribute to the gain. We designed an experiment to roughly verify this intuition where we make a student model learn from its own RE (i.e., supervision provided but not knowledge from a powerful teacher). We observed that the performance ordering is REKD(teacher_distillation) >> self_distillation > RE for both ViT Small and Tiny on CIFAR10, which verifies that both (1) and (2) contribute, but (1) is more influential.
>
>    For ViT Small (Acc@R%): REKD is .968@14.8; Self_Distil is .911@15.1; RE is .889@14.9.
>
>    For ViT Tiny: REKD is .936@14.9; Self_Distil is .82@15; RE is .797@14.8.
>
> 2. Teacher seed: While standard practice in KD is to freeze a single teacher across repeated student runs, we appreciate the reviewer's point on teacher rationale quality. To rigorously test this, we evaluated REKD using paired random seeds (where student seed M learns from teacher seed M). These new experiments on CIFAR 10, reported below, confirm that the gains vs. RE are robust to the teacher seed.
>
>    For ViT Small (Acc@R%): Paired_seed is .965@15.1; RE is .889@14.9.
>
>    For ViT Tiny: Paired_seed is .926@15; RE is .797@14.8.
>
> 3. Separate distillation: See experiments in Weakness 1(a) response to Reviewer BzfV. Results confirm that distilling rationale brings most of the gain.
>
> 4. Significance test on BERT results: We have conducted paired student’s t-tests on RE and REKD’s IMDB accuracy values. For BERT Small and Mini, we separately have P-value = 0.0000002431<0.05 and 0.0000000686<0.05, showing that REKD accuracy is significantly higher than RE’s.
>
> 5. Against human annotation alignment: Our argument in this paper, while being concise, has strong analysis, which could serve as a starting point for a comprehensive position paper in the future. We will acknowledge this in both limitation and future work.
>
> 6. Training cost: KD aims to lower cost for inference, not training. Comparing a smaller student to a larger student would not be fair for a KD method as they differ in inference cost.
>
> 7. Datasets/models: The used ViTs were pretrained on ImageNet as the un-pretrained ViTs take hundreds of epochs to converge on CIFAR data which is too expensive. Note that each RE model has two base neural nets (e.g., the BERT Base RE model has ~2*110M params). More experiments on larger scale datasets/neural models are desirable but might be infeasible with the limited time of this rebuttal.
>
> **Question responses:**
>
> 1. Seeds: See Weakness 2 response
>
> 2. Separate distillation: See Weakness 3 response
>
> 3. Rationale quality: More comprehensive human confirmation over the REKD vs. RE rationales to support the value of the improved rationales is interesting but is likely too much for the timetable for this rebuttal. It will be acknowledged in limitation.
>
> 4. Architecture-agnostic: The architecture-agnostic nature of REKD is guaranteed by design. While we agree with the reviewer that the evaluated variants belong to the same architectural families, they still present the challenge of different hidden dimensions for feature representations (e.g., BERT Base dim=768 vs. Small dim=512), which REKD successfully avoids. Distilling across different architecture families introduces differing inductive biases; this poses a challenge that is worth both empirical and theoretical investigation in future work. We will acknowledge the missing cross-architecture distillation in limitation.
>
> 5. BERT REKD outperforming CLS: We agree that it is worth more discussion. It is a good example against the belief on the interpretability and accuracy trade-off. One explanation is that the teacher model delivers rationales with strong predictive patterns that are not well learnt by the black box student CLS model. One way to think about it is that humans could predict an object’s trajectory by following Newton’s laws more precisely than the intuitive knowledge encoded in their complex biological neural nets. Overall, it depends on the dataset and the models.
>
> 6. Dataset generalization: The popular datasets and models used actually greatly saved experimental costs as we already have decent knowledge on the teacher and student models’ capability gaps on the datasets, which helps REKD experiments. In harder datasets, the teacher’s rationales, even degraded, should still be better than the student’s which might be more degraded. We can also turn to a more powerful teacher if needed. What mainly matters might be the capability gap and, again, it is empirical. We will acknowledge that investigating REKD further on more complex datasets is interesting.
>
> **Limitation to include:** As suggested, we will acknowledge these points in limitations. We will also reflect the added experiments in the paper.

---

> > ### Author Rebuttal · Reviewer_iSGE · 2026-04-03
> >
> > I appreciate the authors' thorough responses and the additional experiments they ran - the ablations and seed robustness checks genuinely strengthen the paper. The results are impressive, especially the gains for ViT-Tiny and BERT-Mini, and I'm holding my score at 4. That said, my main hesitation is that CIFAR and IMDB are pretty easy benchmarks by 2026 standards, and it's hard to know if the method holds up at the scale where lightweight interpretable models would actually be deployed. The "model-agnostic" claim is also only tested within the same architecture families, which feels like an important gap given how central that framing is. I'd love to see a revised version with harder benchmarks and a more rigorous treatment of rationale quality beyond just accuracy.

---

> > > ### Author Response · Authors · 2026-04-04
> > >
> > > We thank the reviewer for the feedback and the concern.
> > >
> > > 1. Regarding **datasets and models**: The primary objective of our paper is to establish the general viability of the REKD framework. To this end, we provided comprehensive empirical evidence across two distinct modalities, utilizing multiple popular variants of both BERTs (for language tasks) and ViTs (for vision tasks). While these datasets may be easily solved by massive, dense models today, they remain highly challenging for lightweight models, when forced to predict through a highly sparse rationale bottleneck. Our experiments demonstrate that standard RE suffers severe performance degradation in these constrained settings, whereas REKD successfully recovers predictive accuracy.
> > >
> > >    REKD is designed to be inherently model-agnostic. Because it relies on a universal interface of feature selection (the rationale mask) that is decoupled from the internal architecture of the base neural networks, the framework can theoretically be applied to any architecture that share the same input features. For example, REKD has been experimented on BERTs and ViTs where teacher and student have different hidden dimensions for feature representations.
> > >
> > >    While we agree with the reviewer that evaluating REKD further on more complex datasets and different neural families (e.g., ViT-ResNet) would further showcase its versatility, we believe our current cross-modal experiments sufficiently validate the core claims of the paper. We will explicitly acknowledge these broader applications and datasets as important avenues for future work in the revised manuscript.
> > >
> > > 2. Regarding **rationale quality**: Because the predictor in an RE framework relies exclusively on the selected features, predictive performance serves as a direct proxy for the quality of the rationales. Our results demonstrate a significant improvement in RE predictive performance of a student model, and our ablation studies explicitly isolate the positive effects of teacher-rationale distillation and stable training from supervised learning (by self-rationale distillation). Together, these show that a student model successfully learns from a teacher's predictively powerful rationales to improve its own feature selection, directly leading to better predictions. Facilitating this transfer of rationale quality is the core contribution of REKD.
> > >
> > >    Unlike human annotation alignment metrics (which can unfairly penalize a faithful model for highlighting genuine dataset biases as we have discussed in Section 3.4), the evaluation on predictive performance under a rationale constraint provides an objective evaluation of rationale quality. The difficulty of evaluating RE models on real world datasets (like ours) is that we do not have ground-truth feature maps for statistically predictive features. In our response to **hZs6** Question 1, we designed a synthetic dataset where we could control the importance of the features. The experiment showed that RE could learn rationales that were truly informative.
> > >
> > >    Ultimately, once a high-capacity RE teacher is validated to produce high-quality rationales (irrespective of the specific quality metric a practitioner chooses to employ), REKD could utilize those teacher rationales to supervise a student RE model. The knowledge distillation in REKD acts as a strong regularization on the student's feature selection, yielding improved rationale quality.

---

### Official Review · Reviewer_hZs6 · 2026-03-13

**Soundness:** 3
**Presentation:** 3
**Significance:** 2
**Originality:** 2
**Overall Recommendation:** 3
**Confidence:** 3

**Summary:**

The paper proposes REKD, a knowledge distillation method for rationale extraction (RE) models. In addition to the standard RE objective, student models are trained to match the teacher’s rationale and prediction distributions. Experiments on vision (CIFAR-10/100 with ViTs) and language (IMDB with BERT variants) show consistent improvements for smaller models trained with REKD compared to standard RE training.

**Compliance With Llm Reviewing Policy:**

Affirmed.

**Final Justification:**

The authors have answered my questions. However, I will maintain my initial rating due to limitations in the experimental evaluation (no human evaluation, various issues mentioned by other reviewers).

**Key Questions For Authors:**

Can you think of experiments that would help validate that the selected rationales reflect informative input features?
Could evaluations on synthetic datasets with ground-truth feature maps help?

**Limitations:**

yes

**Strengths And Weaknesses:**

**Strengths**
1. **Straightforward approach:** The method is well motivated and technically coherent. The architectural independence between teacher and student models is a good feature.
2. **Strong experimental section:** The experimental setup clearly highlights the performance gains of REKD, and the cross-modality evaluation strengthens this. Results are easy to interpret, well explained, and enough detail is provided for reproducibility.
3. **Clear writing:** The paper is generally well written and understandable.

**Weaknesses**
1. **Human understandability and qualitative analysis:** While the authors argue that alignment with human intuition is not a necessary evaluation criterion, rationale extraction is ultimately motivated by enabling better human understanding of model behavior. The paper would benefit from additional qualitative examples illustrating what kinds of features are selected and how REKD qualitatively differs from standard RE.
2. **Faithfulness and grounding of selected features:** The evaluation focuses on predictive accuracy under sparsity constraints. Although predictions are necessarily based on the selected features, it is unclear whether the selected rationales consistently reflect informative structure in the input, or whether the generator and predictor might learn sparse codes in which the rationale mask serves as a communication channel, as e.g. discussed in [1]. As the paper in its current form does not contain human evaluations that may safeguard against these kinds of problems, additional experiments or discussion addressing this concern would strengthen the interpretability claims of the work.

[1]: Wäldchen, S., Sharma, K., Turan, B., Zimmer, M. &amp; Pokutta, S.. (2024). Interpretability Guarantees with Merlin-Arthur Classifiers. URL: https://proceedings.mlr.press/v238/waldchen24a.html.

---

> ### Author Rebuttal · Authors · 2026-03-31
>
> We thank the reviewer for the valuable feedback.
>
> **Weakness responses:**
>
> 1. Comprehensive human confirmation over the REKD vs. RE rationales to support the value of the improved rationales is interesting but is likely too much for the timetable for this rebuttal. However, we will provide examples in the paper to illustrate the guidance of a teacher’s rationale over a student’s selection.
>
> 2. We acknowledge that the reviewer raises an important point, discussed in [1], regarding the risk of the generator and predictor colluding to use the rationale mask as a communication channel. We would like to argue with two points: (1) structural faithfulness as a diagnostic tool and (2) the distillation in REKD helping to prevent such communication channels.
>
>    (1) In defense of RE, even if a communication channel were to form, the inherently faithful nature of the select-predict architecture ensures interpretability is maintained. If the model turns to passing a 'sparse code' rather than 'informative structures', the resulting rationale will appear not reasonable to a human. This faithfully exposes the model's underlying knowledge, informing the human that the prediction relies on spurious patterns and should not be trusted.
>
>    (2) More importantly, the distillation in REKD could help to prevent this communication channel from forming in the first place. In standard RE, when training a generator and a predictor jointly from scratch, they could co-adapt and create arbitrary sparse codes. By contrast, assuming there is a teacher that learns informative structures, our method could distill the robust, pre-established feature selections of a teacher model into the student since KD performs as a strong regularization over the student’s behavior.
>
>    In short, the work, such as [1], on preventing the forming of such communication channels in rationale extraction is orthogonal but complementary to our work on REKD and the two types of techniques could be utilized together to create stronger students. In the paper, we will reflect on this discussion and the work on preventing sparse codes as communication channels.
>
> [1]: Wäldchen, S., Sharma, K., Turan, B., Zimmer, M. & Pokutta, S.. (2024). Interpretability Guarantees with Merlin-Arthur Classifiers. URL: https://proceedings.mlr.press/v238/waldchen24a.html.
>
> **Question response:**
>
> 1. We completely agree that experiments on a synthetic dataset with ground-truth feature maps are a good way to verify feature selection. Yes, we have tested RE on a synthetic dataset for a linear regression task. In our preliminary experiment, we created a tabular synthetic dataset with 5 features. The features were either binary or real numbers following Bernoulli or Normal distributions. By summing up the feature selection frequencies across all the sample runs, we observed that the ranking of selection frequencies for features followed the weights of features that were used to create the synthetic dataset. It helped to verify that the RE model’s selections could be truly informative.

---

> > ### Author Rebuttal · Reviewer_hZs6 · 2026-04-08
> >
> > The authors have answered my questions. However, I will maintain my initial rating due to limitations in the experimental evaluation (no human evaluation, various issues mentioned by other reviewers).

---

> > > ### Author Response · Authors · 2026-04-08
> > >
> > > Thank you for taking the time to read our rebuttal and for confirming that we have successfully answered your questions.
> > >
> > > Regarding human evaluation, we would like to briefly refer back to the evaluation section of our paper (**Section 3.4**), where we explicitly discuss the limitations of human evaluation for interpretable machine learning. As we highlight in the text, human annotation alignment metrics can be inherently flawed for RE, because they often unfairly penalize a structurally faithful RE model that correctly exposes genuine dataset biases rather than aligning with human logic. As an example, an RE model could learn from a biased dataset that a hospital's name is highly correlated to disease diagnostic predictions where physicians would naturally select medical symptoms as rationales (i.e., misalignment). However, this does not mean the hospital name is bad as a rationale.
> > >
> > > Because of this, we intentionally prioritized the objective metric of predictive performance under rationale length constraints. Nevertheless, we recognize that human-centric usability studies remain a valuable step for real-world deployment and the potential for future human-usability studies will be highlighted in the revised manuscript.
> > >
> > > Thank you again for your valuable feedback throughout this reviewing process.

---

### Decision · Program_Chairs · 2026-04-30

**Decision:**

Accept (regular)

**Comment:**

This paper proposes REKD, a knowledge distillation method for rationale extraction (RE) models. In addition to the standard RE objective, student models are trained to match the teacher’s rationale and prediction distributions. Experiments on vision (CIFAR-10/100 with ViTs) and language (IMDB with BERT variants) show consistent improvements for smaller models trained with REKD compared to standard RE training.

Reviewers commended the paper for its simple methodology and strong experiment results. However, multiple reviewers are stuck about whether human evaluation is necessary in this case. Thus this paper received mixed reviews. The AC holds the belief that human evaluation is not necessary to establish faithfulness and sometimes may even harm the establishment of faithfulness. Thus, for evaluating heatmaps human evaluations should NOT be required. With this major issue lifted, AC suggests to accept this paper.